# Prediction of Postoperative Creatinine Levels by Artificial Intelligence after Partial Nephrectomy

**DOI:** 10.3390/medicina59081402

**Published:** 2023-07-31

**Authors:** Tae Young Shin, Hyunho Han, Hyun-Seok Min, Hyungjoo Cho, Seonggyun Kim, Sung Yul Park, Hyung Joon Kim, Jung Hoon Kim, Yong Seong Lee

**Affiliations:** 1Synergy A.I. Co., Ltd., Seoul 07985, Republic of Korea; shinergizer@gmail.com; 2Department of Urology, College of Medicine, Hallym University, Chuncheon 24253, Republic of Korea; tjdrbs7358@gmail.com; 3Department of Urology, College of Medicine, Ewha Womans University, Seoul 07985, Republic of Korea; 4Department of Urology, College of Medicine, Yonsei University, Seoul 03722, Republic of Korea; tintal@yuhs.ac; 5Tomocube, Inc., Daejeon 34109, Republic of Korea; min6284@gmail.com (H.-S.M.); phelahab@gmail.com (H.C.); 6Department of Urology, College of Medicine, Hanyang University, Seoul 04763, Republic of Korea; syparkuro@hanyang.ac.kr; 7Department of Urology, College of Medicine, Konyang University, Daejeon 35365, Republic of Korea; hjkim@kyuh.ac.kr; 8Department of Urology, Chung-Ang University Gwangmyeong Hospital, College of Medicine, Chung-Ang University, Gwangmyeong 14353, Republic of Korea; uroking@cauhs.or.kr

**Keywords:** artificial intelligence, acute kidney injury, postoperative renal function, partial nephrectomy

## Abstract

*Background and Objectives*: Multiple factors are associated with postoperative functional outcomes, such as acute kidney injury (AKI), following partial nephrectomy (PN). The pre-, peri-, and postoperative factors are heavily intertwined and change dynamically, making it difficult to predict postoperative renal function. Therefore, we aimed to build an artificial intelligence (AI) model that utilizes perioperative factors to predict residual renal function and incidence of AKI following PN. *Methods and Materials*: This retrospective study included 785 patients (training set 706, test set 79) from six tertiary referral centers who underwent open or robotic PN. Forty-four perioperative features were used as inputs to train the AI prediction model. XG-Boost and genetic algorithms were used for the final model selection and to determine feature importance. The primary outcome measure was immediate postoperative serum creatinine (Cr) level. The secondary outcome was the incidence of AKI (estimated glomerular filtration rate (eGFR) < 60 mL/h). The average difference between the true and predicted serum Cr levels was considered the mean absolute error (MAE) and was used as a model evaluation parameter. *Results*: An AI model for predicting immediate postoperative serum Cr levels was selected from 2000 candidates by providing the lowest MAE (0.03 mg/dL). The model-predicted immediate postoperative serum Cr levels correlated closely with the measured values (R^2^ = 0.9669). The sensitivity and specificity of the model for predicting AKI were 85.5% and 99.7% in the training set, and 100.0% and 100.0% in the test set, respectively. The limitations of this study included its retrospective design. *Conclusions*: Our AI model successfully predicted accurate serum Cr levels and the likelihood of AKI. The accuracy of our model suggests that personalized guidelines to optimize multidisciplinary plans involving pre- and postoperative care need to be developed.

## 1. Introduction

Partial nephrectomy (PN) is the standard treatment for small renal masses when malignancy is suspected [1]. Even for larger anatomically complex renal tumors, PN is increasingly performed using newly developed surgical techniques such as surgical robots [2]. Renal function is reported to decline after PN by approximately 10% immediately after surgery, recover to some degree, and stabilize after three months [3,4]. Renal function is monitored by estimated glomerular filtration rate (eGFR, mL/min/1.73 m^2^), which is based on serum creatinine (Cr) levels and is adjusted by age, sex, and ethnicity [5].

Acute kidney injury (AKI) is a potentially life-threatening adverse event of PN [6], with incidence rates as high as 30% [7]. AKI may lead to a chronic state or even to irreversible end-stage renal disease (ESRD) [8]. A nationwide survey estimated the annual cost of inpatient care related to AKI itself as over 1% of the total medical expenses [9], not considering the larger socioeconomic burden of ESRD. Immediate postoperative serum Cr levels are considered a better indicator of long-term renal function than those at later time periods, which can be influenced by other nonsurgical events/interventions (i.e., renal-toxic drugs and infection) [10].

Factors associated with AKI following PN include medical factors (age, presence of a solitary kidney, diabetes, hypertension, preoperative eGFR) and surgical factors (estimated blood loss, anesthesia time, and warm ischemic time) [4,6,7,11]. For PN, the anatomical complexity of the tumor is also an important factor of postoperative renal function, which can be quantified by using nomograms, such as the preoperative aspects and dimensions used for anatomical classification (PADUA) or the radius, exophytic/endophytic properties, nearness of the deepest portion of the tumor to the collecting system or sinus, anterior/posterior descriptor, and the location relative to the polar lines (RENAL) nephrometry score [12,13]. In addition, the amount of functional nephron loss by tissue resection (excluding the tumor itself) and the ischemia counts for the reduction of affected kidney function are important measurements [14]. Therefore, it is challenging to predict postoperative renal function and the onset, severity, and duration of AKI, given the numerous preoperative and intraoperative factors affecting renal function and their complex interactions.

The three aspects of post-PN AKI risk factors (medical, surgical, and tumor) have complex interactions [15]. In addition, AKI is a dynamic event with a spectrum of severities that further complicate the formation of a prediction model [16]. In this regard, we hypothesized that the data could be best assessed by artificial intelligence (AI) because of its ability to consider many factors and the relative speed of these approaches. Recently, the use of artificial intelligence has revolutionized the field of healthcare by enabling accurate diagnoses and predictive assessments of diverse ailments [17,18]. A machine learning (ML)-based model was developed to predict AKI in patients with solitary kidneys who underwent PN [19]. However, no prior studies have utilized AI to predict postoperative serum Cr levels, which then predict AKI, determining its severity as well as “near-miss” events. Herein, we report a model for predicting postoperative serum Cr levels and examine the possibility of AKI using AI for technological innovations in PN patient care.

## 2. Methods

### 2.1. Study Design and Population

This study was approved by the Ethics Committee of Ewha Womans University Medical Center (approval No. 2022-08-028) and was conducted in compliance with the Declaration of Helsinki. The data were collected from a customized database and analyzed. All methods were performed in accordance with the relevant institutional guidelines and regulations. This study was conducted using data from patients who underwent PN for suspected kidney cancer based on contrast-enhanced computed tomography (CT) scans and were followed up for longer than one year regarding renal function and oncologic outcome. Patients treated between May 2006 and May 2019 at participating institutions were screened. The inclusion criteria were as follows: (1) patients who underwent PN for a solid enhancing renal mass suspicious for renal cell carcinoma (cT1, ≤7 cm) diagnosed by contrast-enhanced CT; and (2) patients who had preoperative and postoperative serum Cr levels and eGFR. The exclusion criteria were as follows: (1) patients with a single kidney; (2) patients with two or more tumors in one or both kidneys; and (3) patients without renal function follow-up data for 12 months before and after surgery. PN was performed according to the same protocol, including patient selection for comorbidities, surgical methods, and postoperative patient care at the participating institutions. Finally, the imaging data and medical records from participating institutions were analyzed in 785 patients, and the analyzed data were de-identified and transferred to the Core Laboratory located at Ewha Womans University Medical Center for data integration and model development.

### 2.2. Data Collection

#### 2.2.1. Preoperative Features

Age, sex, height, weight, preoperative blood urea nitrogen level, and serum Cr level were retrieved from the Electronic Medical Record (EMR). eGFR was calculated based on data obtained from the EMR using the abbreviated Modification of Diet in Renal Disease (MDRD) equation. In this study, AKI was defined by the Kidney Disease: Improving Global Outcomes (KDIGO) criteria as follows [20]: (1) an increase in serum Cr by 0.3 mg/dL or more within 48 h and (2) an increase in serum Cr to 1.5 times the baseline or more within the last seven days. Because of the retrospective nature of the study, the third criterion, urine output of less than 0.5 mL/kg/h for six hours, was not used. AKI severity was defined as follows: stage 1, serum Cr 1.5–1.9 times the baseline or a ≥0.3 mg/dL increase; stage 2, 2–2.9 times the baseline; stage 3, an increase in serum Cr to ≥4 mg/dL or the initiation of renal replacement therapy. The anatomical complexity of the tumor was determined by three urologists reviewing the preoperative contrast-enhanced CT images using the RENAL nephrometry score, PADUA classification, and centrality index (C-index) [12,13,21], with a slight recent modification to the PADUA system [22]. Resected and ischemic volumes (RAIVs) of the kidneys were calculated as previously described [14].

#### 2.2.2. Intraoperative Features

The type of surgery (open vs. robotic), estimated blood loss (EBL), total anesthesia time (TAT), total operation time (TOT), and warm ischemia time (WIT) data were retrieved from EMR and operation videos.

### 2.3. Model Development

We developed an ML model (SYN-PRF-AN. v1.0.0; Synergy A.I. Co., Ltd., Seoul, Republic of Korea) to predict postoperative Cr levels using the XgBoost 1.7.3 (eXtreme Gradient Boosting) algorithm [23], a gradient-descent algorithm used to search a group of candidate solutions to find the most effective one. Using the aforementioned features (a total of 44 features), the ML model was trained for the levels of immediate postoperative serum Cr. The mean absolute error (MAE) was calculated by subtracting the model-derived Cr value from the individual ground true value and was used as a reference for each model’s fitness. Given the variety of forms and distribution of the data collected, we used a genetic algorithm as a feature selection method, which selected the best “performing” subset from the whole feature set by reinforcement learning. The performance was determined by the model fitness, herein, MAE. We trained and validated our model with a 5-fold cross-validation scheme, and the accuracy of the model was defined by averaging the accuracies over the five tests performed in multiple rounds of cross-validation. The means and confidence intervals were calculated from these multiple trials, where the models were trained on a randomly split train-validation set. Among these models, we selected the model with the best performance (the model with the lowest MAE) and tested it using the hold-out test dataset. By addressing the threats posed by missing values, imbalanced data, and the risk of overfitting, we aimed to ensure the reliability and generalizability of our ML model’s performance in predicting postoperative Cr levels.

### 2.4. Outcome Measures

The primary endpoint was the difference between the model-predicted and the actual immediate postoperative serum Cr levels. The secondary endpoint was the difference between the predicted and actual incidences of AKI events. In addition, we assessed the frequency of the features selected from all the input data using the models and the correlation among the features included in the final model.

### 2.5. Statistical Analysis

Demographic and perioperative data are presented as descriptive statistics. The count data are expressed as percentages, and continuous data are presented as means and standard deviations. A Bland–Altman plot (difference plot) was used to analyze the agreement between the observed Cr values and model-predicted values. Student’s *t* tests were used to compare the differences between the measured and model-predicted values. For all statistical analyses, two-sided *p*-values < 0.05 were considered to indicate statistical significance. All analyses were conducted using *R* [24].

## 3. Results

The initially collected data consisted of renal function follow-up data of 794 PN cases for one year before and after surgery. After excluding nine cases with the exclusion criteria, a training dataset (n = 706) and a test dataset (n = 79) were generated (Table 1). In this study, 44 features known to be associated with postoperative renal function were selected as inputs, including TAT, type of surgery, preoperative serum Cr levels, and eGFR. From the original training set, we created 2000 new training sets (T1~T2000, each n = 706) by sampling using the replacement method. An independent prediction model was generated (M1 ~ M2000). The performance of each model was evaluated by comparing the MAE of the predicted value with the ground truth Cr value (Figure 1).

### 3.1. Feature Importance

The most commonly selected features in the model training stage were preoperative eGFR, sex, and TAT (Figure 2a). The model with the lowest MAE in this study used the following features: age, sex, height, tumor size, TAT, type of surgery, preoperative eGFR and Cr levels, polar location, renal rim, sinus, and collecting system of PADUA scoring (bold font in Figure 2a). We generated a Pearson correlation matrix of the features used in the model with the lowest MAE (Figure 2b). The radiological features showed very high intercorrelations, whereas the remaining clinical features were independent variables.

### 3.2. Model Performance

The Cr values predicted by the model correlated very well with the ground truth Cr values (R^2^ = 0.9669) (Figure 2c). The MAE of the predicted serum Cr level of the model was 0.034 mg/dL. The distribution of the differences between the ground truth and predicted values was similar to a normal distribution (Figure 2d). We compared the performance of the models in predicting the development of AKI (postoperative eGFR < 60 mL/min). The MAE of the AKI group was 0.038 mg/dL, which was not significantly different from that of the whole population (0.034 mg/dL).

### 3.3. AKI Prediction

In the training set (n = 706), 69 (9.8%) AKI events occurred in the immediate postoperative period, as defined by the KDIGO criteria. Two events (2.8%) were stage 2 (serum Cr 2–2.9 times the baseline), and the rest were stage 1 (serum Cr 1.5–1.9 times baseline or ≥0.3 mg/dL increase). In the test set (n = 79), there were three (3.8%) AKI events, all stage 1. The sensitivity and specificity of the model for predicting AKI were 85.5% and 99.7% in the training set, and 100.0% and 100.0% in the test set, respectively (Figure 3a,b). Notably, AKI staging predictions were 100% accurate (Figure 3c,d).

### 3.4. Clinical Scenario

Given that the other factors in the model (age, sex, height, tumor size, type of surgery, preoperative eGFR and Cr levels, and PADUA score) were fixed, we drew a 2D projection plot of the predicted serum Cr levels by TAT as an adjustable feature (Figure 3e). The model implied that (1) the relationship between postoperative Cr levels and TAT was nonlinear, and (2) Cr levels were predicted to be stable when TAT < 150 min and increased when TAT exceeded 150 min.

## 4. Discussion

Precision medicine, or predictive, preventive, personalized, and participatory (4P) medicine, refers to individualized disease prevention and treatment. The average often fails to represent the characteristics of a diverse or heterogenous group of individuals. However, owing to technical limitations, with current evidence-based medicine, the average values or overall trends of the target disease group are used to guide clinical decisions. Recently, owing to innovations in the field of data science, more personalized analyses of each patient are possible with large-scale genomic, proteomic, metabolomic, and clinical databases. In this context, an AI-based approach has the potential to fully utilize the rich data and enable true 4P medicine to be provided [25]. The use of AI in diagnosing and predicting the prognosis of various diseases has indeed been extensively explored in medical research [26,27,28,29,30,31,32]. In the clinical field of urology, more than 100 papers have been published on the use of AI in clinical applications, such as in the diagnosis of various urological malignancies in pathology, surgical outcomes of various cancers and urolithiasis, treatment planning for radiotherapy, and drug selection [16,33].

Predicting the likelihood and severity of AKI after PN is challenging given the numerous preoperative and intraoperative factors affecting renal function and their complex interactions. Therefore, we used AI to build a model using patient age, sex, height, preoperative serum Cr and eGFR, type of surgery, renal tumor size, and complexity factors to cooperatively predict Cr levels immediately after surgery. While previous studies have shown that AKI events themselves can be predicted with acceptable accuracy, our model provides the advantages of predicting the Cr levels and eGFR directly, estimating the severity of AKI as low as a subclinical level of injury, and can be interpreted contextually.

Residual nephron volume and various other factors, such as age, preoperative renal function, operation time, and tumor location (RENAL nephrometry score), can affect postoperative outcomes [6]. Nomograms using conventional statistical methods predict the likelihood of postoperative AKI by combining preoperative medical or surgical factors [11,34]. Using these methods, residual renal function after PN could be predicted based only on statistical trends, not at the individual patient level. In contrast, our model predicts Cr levels on the immediate postoperative day for each individual, providing an exact eGFR and accurately predicting AKI occurrence and severity. The statistical methodology that we have used so far predicts the probability of AKI at a level of a few percent, so it is difficult to provide specificity in how to prepare for that risk. However, the predicted Cr value provided by our AI model is derived from the stochastic possibility and might be helpful for actual clinical treatment, enabling a practical approach to decide the level of operation and anesthesia to be performed and the management of various risk factors. Thus, our AI model might maximize the chance of detecting AKI and administering interventions in a timely manner, which may save patients with chronic kidney disease [10].

One of the advances in AKI patient care is the introduction of an electronic alert (e-alert) system. The e-alert system reduces the real-world response time to changes in Cr levels and the chances of further renal impairment [35]. Our model can be integrated into an e-alert system that identifies patients at risk of acute damage. For example, we present a clinical scenario in which the TAT threshold can be used to predict the occurrence of postoperative AKI. While it is generally consistent with previous reports that eGFR decreases during the duration of the surgical procedure [7], our model further indicates an inflection point of TAT and AKI risk relationship. This information might be helpful in the operating room to customize the treatment strategies to avoid AKI. One plausible explanation for this observed dependency lies in the potential impact of prolonged anesthesia and surgical procedures on renal function. During lengthier operations, a cascade of factors can come into play, including reduced renal blood flow, compromised renal perfusion, and the occurrence of ischemia–reperfusion injury [36]. These mechanisms can collectively contribute to postoperative kidney dysfunction, subsequently leading to elevated creatinine levels. Consequently, a longer total anesthesia time indirectly affects renal function and ultimately influences the changes observed in postoperative creatinine levels.

It is important, however, to acknowledge the intricacies inherent in the relationship between total anesthesia time and postoperative creatinine elevation. This relationship is multifaceted and influenced by a multitude of patient-specific factors, such as baseline renal function, comorbidities, and surgical intricacies. Furthermore, various surgical techniques and perioperative management practices can also contribute to the overall impact on renal function and subsequent creatinine level alterations.

Notably, the PADUA system was most often selected between the RENAL and PADUA scoring systems. Previous comparative studies have demonstrated that the RENAL and PADUA scores have nearly equal predictive power [37,38]. However, in our study, PADUA scoring components regarding renal rim, renal sinus, and urinary collecting system (UCS) involvement seemed to be preferred by AI to enhance prediction performance. It might be relevant to refer to the components of a recently modified version of PADUA, the Simplified PADUA REnal (SPARE) classification [22]. Since the SPARE system no longer uses UCS involvement, we combined renal sinus involvement and UCS involvement components and scored them as 0, 1, 2, or 3 (absent, UCS only, sinus only, or both, respectively). Further studies are necessary to validate our strategy of using the modified PADUA system.

A strength of our study is that we used a multi-institutional dataset of intraoperative variables, and baseline features were not associated with the treatment outcomes. These advantages enabled us to overcome the frequent “overfitting” issues encountered in AI-based modeling in the clinical fields [39]. In general, all available inputs should be used to build an ML model with the best performance, and the importance of features may be “interpreted” afterward. In this study, we selected data inputs that have been associated with postoperative renal function outcomes according to clinical experience and previous studies, which resulted in a model of high compliance despite the relatively small number of features.

However, we recognize that this study has several limitations. First, the study population was relatively small and consisted mostly of Asian patients with limited variation in body mass index. External validation may be necessary for populations from different regions and ethnicities. Second, because this study was retrospective, patients with unfit health conditions (uncontrolled diabetes or hypertension and advanced disease states) were excluded. Therefore, it is difficult to claim that the features studied were associated with postoperative AKI in all the patients. Third, for the association between anesthesia time and postoperative renal function, there are other possible explanations for the inflection points, such as surgical complexity and the amount of functional parenchyma removed or the year of surgery, which were not assessed in this model. Fourth, our dataset presented challenges including missing values, a limited number of samples, and the risk of overfitting. Collecting data from multiple medical centers resulted in missing values, making the dataset prone to evaluation as an imbalanced dataset. Additionally, the limited number of samples increased the likelihood of overfitting, where the model becomes overly complex and fails to generalize well to new data. To mitigate these challenges, we employed a genetic algorithm for feature selection and implemented a 5-fold cross-validation scheme to ensure robust evaluation. By carefully addressing these threats, we aimed to develop a reliable and generalizable ML model for predicting postoperative Cr levels.

In the future, the presented model can be enhanced by various populations with racial differences, hospital conditions, and the addition of our previously developed AI algorithm that directly measures residual nephron volume following PN, as residual nephron volume after surgery has been shown to be useful in predicting the renal function preservation effects of PN in advanced cases of T2 or higher disease [40].

## 5. Conclusions

We found that it is feasible to build a postoperative serum Cr level prediction model using AI with a high accuracy for PN. Rather than dichotomously determining whether AKI will occur, our AI-generated model directly provides immediate postoperative serum Cr levels that can be utilized for timely intervention. Our results further indicate that renal function is best preserved by testing multiple possible factors, such as extended anesthesia time, and they provide multidisciplinary guidance for clinicians. Finally, this study supports AI as an important tool for 4P medicine, where individualized, digitalized patient care is realized, maximizing the socioeconomic output of healthcare.

## Figures and Tables

**Figure 1 medicina-59-01402-f001:**
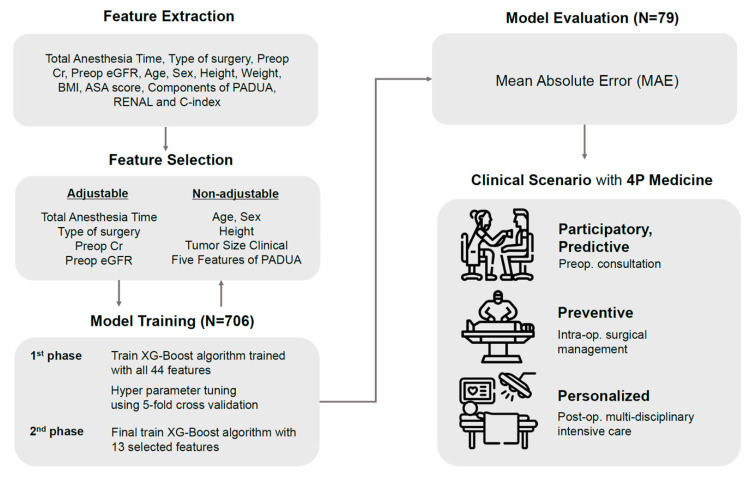
Experimental scheme.

**Figure 2 medicina-59-01402-f002:**
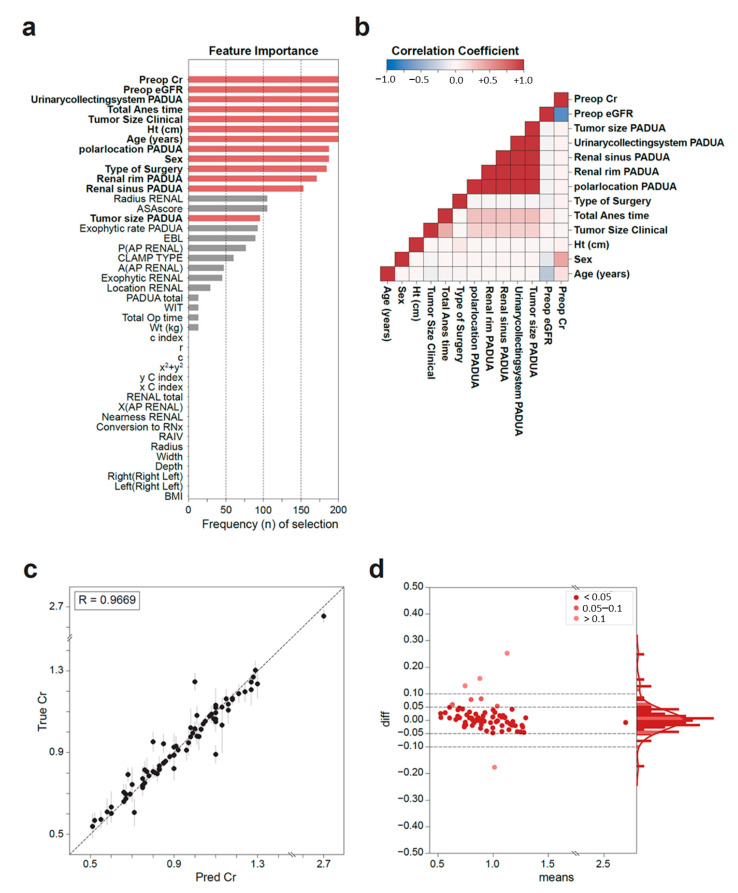
Features selected in the model and model performance. (**a**) Feature importance determined based on the frequency (n) of feature selection in all models. The features are listed by decreasing rank of selection frequency. Features selected by the best-performance model are marked in bold font. (**b**) Correlation matrix of the selected features. (**c**) Scatter plot of the predicted day 0 serum creatinine (Cr, mg/dL) values (X-axis) and ground truth values (Y-axis). The vertical bar indicates the standard error. (**d**) Bland–Altman plot showing the distribution the differences between the predicted and true Cr levels (mg/dL) for all patients. The right axis shows the distribution histogram. The color of the dot indicates the size of the difference (box).

**Figure 3 medicina-59-01402-f003:**
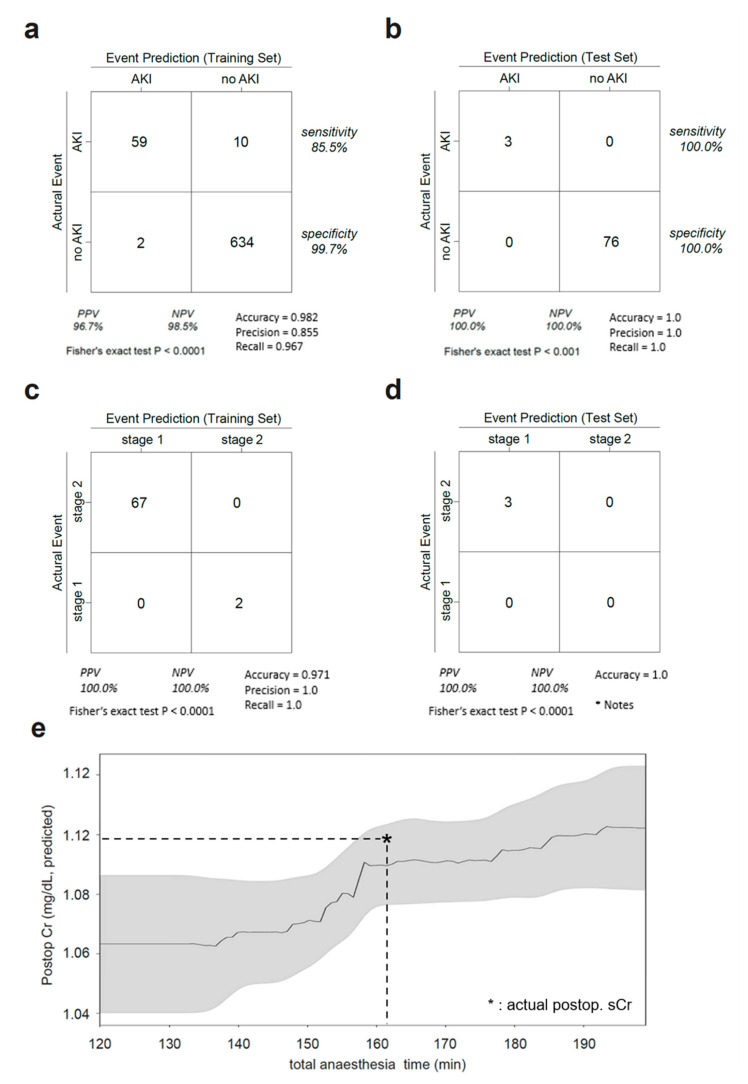
AKI prediction performance. (**a**,**b**) A 2 × 2 table of the predicted and actual AKI events in the training set (**a**) and test set (**b**). PPV, positive predictive value; NPV, negative predictive value. (**c**,**d**) A 2 × 2 table of the predicted and actual AKI events in the training (**c**) and test (**d**) sets. PPV, positive predictive value; NPV, negative predictive value. * Notes: NPV is undefined when there are no True Negatives and False Negatives. Precision and recall cannot be calculated as there are no positive samples in this configuration. (**e**) A sample clinical scenario: postoperative Cr level predicted by the total anesthesia time, patient sex, age, height (Ht), surgery type, tumor size, and PADUA score. In this scenario, Cr levels surged when the anesthesia time increased from 150 to 170 min. The Cr level was predicted to be stable in other time windows.

**Table 1 medicina-59-01402-t001:** Baseline patient characteristics.

		Data Type	Training Set	Test Set	*p*-Value
	Number of Patients		706	79	
Patient Physical Status	Age (years)	n	53.4 ± 12.7	50.2 ± 12.0	0.478
	Sex (Male: Female)	n	461:244	46:32	
	BMI (kg/m^2^)	n	24.6 ± 3.6	25.2 ± 4.35	0.345
	DM (Diabetes Mellitus)	c	136	15	
	HTN (Hypertension)	c	356	36	
	ASA score (1:2:3)	n	1.52 ± 0.77	1.56 ± 0.64	0.642
Tumor Radiology	Location (Left/Right)	c	335/367	38/40	
	Depth (cm)	n	1.74 ± 1.17	1.51 ± 0.94	0.124
	Width (cm)	n	1.09 ± 0.23	1.13 ± 0.24	0.246
	Radius (cm)	n	1.55 ± 0.87	1.34 ± 0.64	0.091
	RAIV, resected volume (cm^3^)	n	35.9 ± 38.5	28.7 ± 20.9	0.062
	RAIV, ischemized volume (cm^3^)	n	58.8 ± 75.9	42.2 ± 38.0	0.059
Surgery	Clamp Type (zero ischemia: selective: full)	c	110: 117: 440	11: 20: 44	
	Total Operation Time (TOT, min)	n	177.0 ± 78.2	164.5 ± 75.6	0.134
	Total Anesthesia Time (TAT, min)	n	245.2 ± 86.3	237.6 ± 72.4	0.541
	Warm Ischemia Time (WIT, min)	n	21.3 ± 12.7	20.3 ± 11.7	0.633
	Type of Surgery (Robot: Open)	n	620: 38	73: 1	
	Estimated Blood Loss (EBL, mL)	n	498.6 ± 715.4	376.3 ± 355.6	0.111
	Conversion to Radical Nephrectomy (True/False)	n	10:427	0:37	
	Sliding clip renorrhapy (n)	c	625	65	
	Use of reno-protective agents	c	34	6	
PADUA Score *	Polar location (upper/lower: medium)	c	407:270	38:39	
	Exophytic rate (≥50%:<50%: endophytic)	c	224:330:123	21:41:15	
	Rim location (lateral: medial)	c	454:223	46:31	
	Renal sinus involvement (absent not: present)	c	400:277	39:38	
	Sinus/UCS involvement (absent: sinus only: UCS only: both)	c	392:277:7:1	38:38:1:0	
	Tumor size (cm) (≤4:4.1–7:>7)	c	516:135:26	60:16:1	
	PADUA total (score)	n	8.70 ± 1.77	9.06 ± 1.78	0.125
RENAL Score	Radius (≤4:>4 but 7:>7)	c	510:136:31	60:16:1	
	Exophytic properties (≥50%:<50%: endophytic)	c	228:326:122	24:39:14	
	Nearness to Sinus or UCS (mm) (≥7:>4 but <7:≤4)	c	249:108:320	22:15:40	
	Location to coronal plane Anterior (A): Posterior (P): Neither (X)	c	328:329:18	38:38:1	
	Location to polar line (upper/lower: cross: across or midline)	c	277:206:194	37:16:24	
	RENAL total (score)	n	6.11 ± 2.11	6.60 ± 2.00	0.411
Centrality Index (C-index)	x (cm)	n	2.04 ± 2.20	2.84 ± 4.65	0.365
	y (cm)	n	2.90 ± 4.96	4.30 ± 7.01	<0.05
	$x^{2}+y^{2}$	n	41.8 ± 232.8	94.6 ± 239.1	<0.05
	*c* $\left(\sqrt{x^{2}+y^{2}}\right)$	n	3.87 ± 5.19	5.78 ± 7.97	<0.05
	*r* (d/2)	n	1.53 ± 0.86	1.36 ± 0.61	0.123
	C-index (score)	n	2.72 ± 3.27	3.20 ± 4.06	0.241
Renal Function	Preop eGFR (mL/min/1.73 m^2^)	n	87.1 ± 26.2	91.8 ± 22.8	0.289
	Preoperative serum Cr (mg/dL)	n	0.90 ± 0.28	0.85 ± 0.25	0.124
	Postoperative day 0 serum Cr (mg/dL)	n	1.01 ± 0.33	0.94 ± 0.28	0.223

BMI—body mass index; DM—diabetes mellitus; HTN—hypertension; ASA—American Society of Anesthesiologists; RAIV—resected and ischemic volume (preoperatively calculated); PADUA—Preoperative aspects and dimensions used for an anatomical classification of renal tumors; UCS—urinary collecting system; RENAL—Radius, exophytic/endophytic, nearness, anterior/posterior, location nephrometry schemes; *—Revised version of PADUA score (the Simplified PADUA REnal nephrometry system) was used.

## Data Availability

The data for this study are available from the corresponding author upon reasonable request.

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
