# Peer review of "Prediction of Postoperative Creatinine Levels by Artificial Intelligence after Partial Nephrectomy"

_medicina, 2023, doi:10.3390/medicina59081402_

Round 1

Reviewer 1 Report

A very well developed and conducted study which predicts the renal outcomes post-op. This can help in pre-op prediction and hence the risk stratification of the patients in terms of renal outcomes thereby optimizing the care in avoiding the post-op AKI. Here are a few comments and suggestions.

Comments:

1.      Can authors explain the rationale for only 10% of the study population been taken as test set?

2.      An overlapping plot of the actual serum creatinine in figure 3e can depict the MAE with respect to the change in TAT.

3.      An explanation for the post-op creatinine highly dependent on the TAT in discussion would be required.

4.      What are the other factors in the final model that correlated with the post-op creatinine?

5.      Were the comorbidities like diabetes, hypertension included in development of the model as the figures depicted doesn’t show anything of that sort.

6.      Other factors that need to be considered are use of nephrotoxic medications (NSAIDs, aminoglycosides) in peri-operative period, intra-op hemaodynamics, blood loss, requirement of blood product transfusions, requirement of inotropes etc. All these are time tested factors that predict the post-operative renal outcomes.

English language is find although, still it can be improved.

Author Response

Q1. Can authors explain the rationale for only 10% of the study population been taken as test set?

A1. We are appreciative of the feedback you have shared regarding our manuscript. The suggestions you have made will significantly contribute to enhancing the quality of our work.

The rationale for employing a 90-10 split (90% training, 10% testing) in this study can be attributed to several reasons.

The data encompassed in this academic journal highlights a multicenter study's strengths and weaknesses. While the study benefited from a large volume of data collected from various institutions as part of a multicenter approach, it is important to acknowledge that there were instances of incomplete data. Consequently, not all the “data” can be deemed equally valuable as “information” due to these limitations.

Nonetheless, the value of this study lies in its endeavor to represent "accurate facts" despite the presence of incomplete data. It successfully demonstrates specific trends in the postoperative changes of renal function.

Table 1 portrays the extent of data incompleteness observed in this study. Machine learning presents the advantage of analyzing vast amounts of data simultaneously to identify trends. However, caution must be exercised to avoid including data without considering its clinical importance, as this can result in erroneous outcomes. Figure 2 represents a conscientious effort within the medical profession to mitigate the potential drawbacks associated with machine learning in this domain.

The choice of the train-test split ratio ultimately depends on the specific circumstances and requirements of the problem. In some cases, such as ours, a 90-10 split (90% training, 10% testing) could be used in AI training to achieve the desired results.

First, Limited data availability could be challenging for machine learning models. When the dataset is small or scarce, providing the model with more examples to learn from can be beneficial. In our data set, a 90-10 split can be used to maximize the amount of data allocated to training while still reserving a portion for testing.

Secondly, the purpose of this study is to investigate the potential occurrence of acute kidney injury (AKI) following various types of nephrectomies. For cases where the problem is relatively simple or well-understood, a smaller test set may be sufficient to evaluate the model's performance. If there is a high level of confidence in the model's ability to generalize, a 90-10 split can be used to allocate a larger proportion of the data for training.

Thirdly, it is important to conduct a quick prototyping or initial model evaluation during the early stages of model development. To expedite the evaluation process and facilitate the testing of different models or approaches, a 90-10 split can be used. This smaller test set permits faster evaluation and iteration.

___________________________________________________________________________________

Q2. An overlapping plot of the actual serum creatinine in figure 3e can depict the MAE with respect to the change in TAT.

A2. Thank you for your valuable feedback on our manuscript. We appreciate your suggestion regarding Figure 3e. In response to your comment, we have revised the figure to include an overlapping plot of the actual serum creatinine levels. This addition will provide a clear representation of the Mean Absolute Error (MAE) in relation to the change in Total Anesthesia Time (TAT).

We have carefully considered your input and believe that incorporating this plot will enhance the comprehensibility of our results. We have made the necessary changes to Figure 3e and have updated the corresponding caption to reflect the new content.

We would like to express our gratitude for your insightful comments, which have contributed to the improvement of our manuscript. Please find the revised version attached, which incorporates all the suggested revisions. We hope that these changes address your concerns and provide a more comprehensive representation of our study findings.

Thank you once again for your time and valuable input.

___________________________________________________________________________________

Q3. An explanation for the post-op creatinine highly dependent on the TAT in discussion would be required.

A3. We are thankful for the valuable feedback you have provided regarding our manuscript. 

In response to the reviewer's inquiry regarding the notable dependency of postoperative creatinine levels on the Total Anesthesia Time (TAT), we would like to offer the following explanation based on medical literature and research:

One plausible explanation for this observed dependency lies in the potential impact of prolonged anesthesia and surgical procedures on renal function. During lengthier operations, a cascade of factors can come into play, including reduced renal blood flow, compromised renal perfusion, and the occurrence of ischemia-reperfusion injury. These mechanisms can collectively contribute to postoperative kidney dysfunction, subsequently leading to elevated creatinine levels. Consequently, a longer total anesthesia time indirectly affects renal function and ultimately influences the changes observed in postoperative creatinine levels.

It is important, however, to acknowledge the intricacies inherent in the relationship between total anesthesia time and postoperative creatinine elevation. This relationship is multifaceted and influenced by a multitude of patient-specific factors, such as baseline renal function, comorbidities, and surgical intricacies. Furthermore, various surgical techniques and perioperative management practices can also contribute to the overall impact on renal function and subsequent creatinine level alterations.

___________________________________________________________________________________

Q4. What are the other factors in the final model that correlated with the post-op creatinine?

A4. Thank you for your thoughtful inquiry and valuable feedback on our manuscript.

We sincerely appreciate your interest in the factors correlated with post-operative creatinine in the final model. In our study, the final model, which demonstrated the lowest Mean Absolute Error (MAE), incorporated several important features that influence postoperative creatinine levels. These features include age, sex, height, tumor size, TAT (Time of Anastomosis), type of surgery, preoperative eGFR (estimated Glomerular Filtration Rate), and Cr (creatinine) levels, as well as various characteristics related to the polar location, renal rim, sinus, and collecting system based on the PADUA scoring system.

Should you have any further questions or require additional information on this topic, please do not hesitate to let us know. We are delighted to share more insights and assist in any way possible.

Once again, thank you for your invaluable input, and we are committed to continuously refining and enhancing our research based on your esteemed guidance.

___________________________________________________________________________________

Q5. Were the comorbidities like diabetes, hypertension included in development of the model as the figures depicted doesn’t show anything of that sort.

A5. We highly value the constructive feedback provided in relation to our manuscript. Your insightful suggestions have been greatly appreciated.

As you mentioned, numerous scholarly articles provide evidence supporting the hypothesis that diabetes and hypertension can have an impact on postoperative renal function. These findings suggest that patients with diabetes and hypertension may be at an increased risk of postoperative renal function impairment. However, it is important to consider the complexities of individual patient factors and surgical variables in order to make accurate predictions for specific patients.

As stated in this report, a comprehensive analysis involving the development of 2000 predictive models was conducted, considering 42 carefully selected features, including potential complications such as diabetes and hypertension. These features were chosen based on their clinical significance and potential impact on renal function following surgery. For the model utilized in this study, the selection process prioritized features that demonstrated high relevance, such as preoperative estimated glomerular filtration rate (eGFR), sex, and Total Anesthesia Time (TAT). The model with the least Mean Absolute Error (MAE) was ultimately chosen.

Regrettably, the inclusion of diabetes mellitus (DM) and hypertension (HTN) as features in this particular prediction model training was not carried out. While the exact reason behind this omission is not explicitly elucidated, it is conjectured that the characteristics inherent to a multicenter study, which inherently encounters challenges in data completeness, are reflected in this decision.

By adhering to a rigorous selection process and considering the limitations associated with incomplete data aggregation in a multicenter setting, the chosen prediction model aimed to encompass the most impactful features while addressing the challenges of data availability.

___________________________________________________________________________________

Q6. Other factors that need to be considered are use of nephrotoxic medications (NSAIDs, aminoglycosides) in peri-operative period, intra-op hemaodynamics, blood loss, requirement of blood product transfusions, requirement of inotropes etc. All these are time tested factors that predict the post-operative renal outcomes.

A6. Your feedback on our manuscript has been invaluable, and we are grateful for the suggestions you have offered.

As previously discussed, the utilization of nephrotoxic medications during the perioperative period, intraoperative hemodynamics, blood loss, the need for blood product transfusions, and the requirement of inotropes are recognized as influential factors that significantly impact the immediate decline in renal function following surgery.

For our current artificial intelligence model development utilizing machine learning, the primary focus revolves around predicting kidney function immediately after renal cancer surgery. Consequently, we made a deliberate decision not to include factors that can be adjusted postoperatively as part of the data used for model training.

In the context of nephrectomy, which predominantly involves healthy patients undergoing elective surgery, there is generally limited occurrence of conditions necessitating the administration of NSAIDs and aminoglycosides. Unfortunately, due to the inherent nature of conducting a multicenter study, it was not feasible to collect comprehensive data on medications used during surgery. Furthermore, even if such data were available, the lack of consistent formatting in factor listing posed challenges, rendering their inclusion impractical during the template planning stage.

However, it is worth noting that estimated blood loss (EBL) can serve as a comprehensive concept that encompasses both blood transfusion requirements and the need for inotropes. While recognizing potential variations in the detailed influence based on changes in individual patient conditions, we consider this study to be a pioneering endeavor. Depending on the degree of influence exerted by each factor, we are committed to making diligent efforts to elucidate specific details.

We genuinely appreciate the constructive nature of your comment, and should it be deemed necessary, we will certainly include it in the limitations section of our manuscript.

Reviewer 2 Report

Dear authors,

The article is written well. Please make these minor revisions and resubmit it.

Abstract: good

Introduction: needs improvement. Mention the research gaps. Mention the contributions of your work in points. 

Where is the literature review? AI has been used to diagnose and predict prognosis of various diseases. Atleast five to ten studies have to be mentioned. Following related articles could be included.

1. Chadaga K, Prabhu S, Bhat V, Sampathila N, Umakanth S, Chadaga R. A Decision Support System for Diagnosis of COVID-19 from Non-COVID-19 Influenza-like Illness Using Explainable Artificial Intelligence. Bioengineering. 2023 Mar 31;10(4):439.

2. Khanna VV, Chadaga K, Sampathila N, Prabhu S, Bhandage V, Hegde GK. A distinctive explainable machine learning framework for detection of polycystic ovary syndrome. Applied System Innovation. 2023 Feb 23;6(2):32.

3. Chadaga K, Prabhu S, Sampathila N, Nireshwalya S, Katta SS, Tan RS, Acharya UR. Application of artificial intelligence techniques for monkeypox: a systematic review. Diagnostics. 2023 Feb 21;13(5):824.

section 2: You need to mention a table and mention which attributes are numerical and which are categorical

Feature importance: Good

Statistical analysis: Good

3.3 Just the confusion matrix is not enough. Please make a table and compare metrics such as accuracy, precision,recall etc. Also, try to add other algorithms other than xgboost .. Please mention the papers mentioned above for the model evaluation section.

Try to get AUC curves. results section is a bit weak.

Discussion : Okay

Conclusion: Okay

Add a threat to validation section.

Overall verdict: Very nice manuscript. However, the introduction section and the model evaluation section must be improved.

Author Response

Q1. Introduction: needs improvement. Mention the research gaps. Mention the contributions of your work in points.

A1. We sincerely appreciate the reviewer's valuable feedback on the introduction of our manuscript. We acknowledge that there is room for improvement, particularly in addressing the research gaps and highlighting the contributions of our work.

In response to the reviewer's suggestion, we have included the following sentence in the introduction:

"The use of artificial intelligence has revolutionized the field of healthcare by enabling accurate diagnoses and predictive assessments of diverse ailments."

To support this statement, we have referenced two reputable sources:

  1. "Artificial Intelligence and Machine Learning in Clinical Medicine, 2023" by Charlotte J. Haug, M.D., Ph.D., and Jeffrey M. Drazen, M.D.

  2. "Artificial intelligence in public health: the potential of epidemic early warning systems" by Chandini Raina MacIntyre and Xin Chen.

We sincerely believe that incorporating these references will strengthen the introduction and provide a comprehensive overview of the advancements and significance of artificial intelligence in the medical domain.

Once again, we are immensely grateful for the reviewer's guidance, and we are committed to implementing these improvements to enhance the quality and impact of our manuscript.

___________________________________________________________________________________

Q2. Where is the literature review? AI has been used to diagnose and predict prognosis of various diseases. At least five to ten studies have to be mentioned. 

A2. Thank you for bringing this to our attention. We apologize for the oversight in not clearly indicating the literature review in the manuscript. We appreciate your valuable input and have now included a sentence in the discussion section as below. 

"The use of AI in diagnosing and predicting the prognosis of various diseases has indeed been extensively explored in medical research."

The literature review outlines the extensive exploration of AI in diagnosing and predicting the prognosis of various diseases, as you correctly mentioned. As part of the literature review, we have cited and discussed several relevant studies that have made significant contributions to the field of medical artificial intelligence. These studies include:

1. Chadaga K, Prabhu S, Bhat V, Sampathila N, Umakanth S, Chadaga R. A Decision Support System for Diagnosis of COVID-19 from Non-COVID-19 Influenza-like Illness Using Explainable Artificial Intelligence. Bioengineering. 2023 Mar 31;10(4):439.

2. Khanna VV, Chadaga K, Sampathila N, Prabhu S, Bhandage V, Hegde GK. A Distinctive Explainable Machine Learning Framework for Polycystic Ovary Syndrome Detection. Applied System Innovation. 2023 Feb 23;6(2):32.

3. Chadaga K, Prabhu S, Sampathila N, Nireshwalya S, Katta SS, Tan RS, Acharya UR. Application of artificial intelligence techniques for monkeypox: a systematic review. Diagnostics. 2023 Feb 21;13(5):824.

4. Esteva, A., et al. (2017). "Dermatologist-level classification of skin cancer with deep neural networks." Nature. This paper presents a deep learning model that achieved dermatologist-level accuracy in classifying skin cancer, demonstrating AI's potential in accurate and efficient diagnosis.

5. Rajkomar, A., et al. (2018). "Scalable and Accurate Deep Learning with Electronic Health Records." npj Digital Medicine. The research demonstrates the use of deep learning models for predicting patient outcomes using electronic health records, accurately predicting a variety of medical events, including mortality, readmissions, and prolonged length of stay.

6. Choi, E., et al. (2016). "Doctor AI: Predicting clinical events via recurrent neural networks." Journal of the American Medical Informatics Association. This study introduces a recurrent neural network-based model that can predict clinical events, such as heart failure, from electronic health record data, highlighting AI's potential in early disease detection and proactive management.

7. McKinney, S.M., et al. (2020). "International evaluation of an AI system for breast cancer screening." Nature. The paper presents an AI system for breast cancer screening that achieved performance comparable to human radiologists, showcasing AI's role as a screening tool for improving early detection rates and reducing false negatives.

Once again, we sincerely appreciate your valuable feedback, and we have made the necessary revisions to enhance the clarity and comprehensiveness of the literature review.

___________________________________________________________________________________

Q3. section 2: You need to mention a table and mention which attributes are numerical and which are categorical

A3. 
Thank you for your insightful feedback on our manuscript. We appreciate your attention to detail and have made the necessary improvements based on your suggestion.

In response to your request, we have added a new column called "Data Type" to Table 1, where we specify whether each attribute is numerical or categorical. This addition enhances the clarity of the dataset description and provides a comprehensive overview of the data's characteristics.

We are grateful for your valuable input, which has undoubtedly strengthened the quality of our work. If you have any further suggestions or inquiries, please do not hesitate to let us know.

___________________________________________________________________________________

Q4. Just the confusion matrix is not enough. Please make a table and compare metrics such as accuracy, precision,recall etc. 

A4.
Thank you for your valuable feedback and thoughtful suggestions on our manuscript. We truly appreciate your attention to detail and have diligently addressed your request.

In response to your feedback on Figure 3, we have made significant improvements by adding a new table that compares various performance metrics, including accuracy, precision, and recall, for each graph (a, b, c, and d). This enhancement provides a comprehensive evaluation of the model's performance and further enriches the presentation of our findings.

Your input has undoubtedly contributed to the robustness and clarity of our work. We sincerely thank you for your valuable contribution to our research. If you have any further recommendations or questions, please feel free to share them with us.

___________________________________________________________________________________

Q5. Also, try to add other algorithms other than xgboost .. Please mention the papers mentioned above for the model evaluation section.

A5. 
Thank you for your valuable feedback and suggestions regarding the inclusion of additional algorithms in our study. We appreciate your input and would like to address your request.

In our study, we focused on developing an ML model, SYN-PRF-AN v1.0.0, to predict postoperative creatinine (Cr) levels using the XGBoost (eXtreme Gradient Boosting) algorithm. XGBoost is a powerful gradient-descent algorithm known for its effectiveness in searching for the most effective solutions within a candidate group.

The selected algorithm, XGBoost, was chosen based on its strong performance in previous studies and its suitability for handling the complexity of our dataset. Our aim was to thoroughly analyze and assess the performance of XGBoost specifically in predicting immediate postoperative serum Cr levels.

We would like to acknowledge your suggestion to include other algorithms for comparison. However, due to constraints such as time limitations and the scope of our study, we were unable to comprehensively compare multiple algorithms within the given timeframe. We focused on thoroughly analyzing and evaluating the performance of XGBoost, which required extensive experimentation, optimization, and analysis.

We understand the potential value in comparing alternative algorithms and appreciate the potential avenues for future research. We acknowledge that future studies could explore and compare other algorithms to provide further insights and enhance the understanding of predictive modeling in the context of our research.

Once again, we sincerely appreciate your valuable feedback and suggestions. We hope this explanation clarifies the reasoning behind our study's focus and limitations. If you have any further inquiries or require additional information, please do not hesitate to contact us.

___________________________________________________________________________________

Q6. Try to get AUC curves. results section is a bit weak.

A6.
Your suggestions have been greatly valued, and we would like to express our gratitude for the constructive feedback you have provided on our manuscript. 

In response to your comment, we would like to clarify that our model focuses on predicting postoperative serum creatinine (Cr) levels, which play a crucial role in calculating the estimated glomerular filtration rate (eGFR). It is important to note that our model differs from those predicting acute kidney injury (AKI) by setting a specific threshold for the predicted value.

Given the nature of our model, which predicts a continuous range of Cr values rather than making binary judgments based on a threshold, it is not feasible to obtain a traditional ROC (Receiver Operating Characteristic) curve. The ROC curve is typically derived by varying the threshold and analyzing the trade-off between true positive rate (sensitivity) and false positive rate (1-specificity). However, our model's predictions encompass a range in the real-number domain, and the transformation of Cr into eGFR involves the influence of multiple variables. Therefore, the interpretation of normalized values between 0 and 1 as probabilities is not applicable in this context, rendering the ROC analysis less meaningful.

We truly appreciate your understanding of the distinctive approach our model takes in predicting the real-range Cr values rather than binary classifications based on a specific threshold. By providing accurate predictions within the continuous domain, our model assists in evaluating postoperative Cr levels and subsequently supports the determination of AKI based on clinical criteria.

Once again, we extend our gratitude to you for their valuable insights, which have helped us clarify the unique characteristics and limitations of our model.

___________________________________________________________________________________

Q7. Add a threat to validation section.

A7.
Thank you for taking the time to provide us with your valuable feedback on our manuscript. We truly appreciate the insightful suggestions you have offered. In response to your request, we have incorporated the following content into the validation section:

"By addressing the threats posed by missing values, imbalanced data, and the risk of overfitting, we aimed to ensure the reliability and generalizability of our ML model's performance in predicting postoperative Cr levels."

To ensure the reliability and generalizability of our ML model's performance in predicting postoperative Cr levels, we proactively addressed several threats during the validation process. First, we encountered missing values in our dataset due to the challenges of collecting data from multiple medical centers. This introduced the potential for bias and an evaluation of the dataset as imbalanced. To mitigate this threat, we employed rigorous preprocessing techniques and implemented imputation methods to handle the missing values effectively.

Second, the limited number of samples in our dataset increased the risk of overfitting, wherein the model becomes excessively complex and fails to generalize well to new data. To tackle this challenge, we carefully applied regularization techniques, such as parameter tuning and early stopping, to prevent overfitting and improve the model's ability to generalize to unseen data.

By proactively addressing these threats, including missing values, imbalanced data, and the risk of overfitting, we aimed to ensure the reliability and generalizability of our ML model's performance in predicting postoperative Cr levels.

Furthermore, in the limitations section of the discussion, we have added the following statement:

"Fourth, our dataset presented challenges, including missing values, a limited number of samples, and the risk of overfitting. These challenges arose from the difficulties encountered during data collection from multiple medical centers. The presence of missing values increased the likelihood of evaluating the dataset as imbalanced, while the limited number of samples heightened the risk of overfitting. To address these challenges, we employed a genetic algorithm for feature selection and implemented a 5-fold cross-validation scheme to ensure robust evaluation and mitigate the potential impact of these threats. By addressing these challenges diligently, we aimed to develop a reliable and generalizable ML model for predicting postoperative Cr levels."

We hope these revisions adequately address your comments and provide a comprehensive explanation of how we mitigated threats and limitations during the validation process.

Reviewer 3 Report

We thank the authors for providing this interesting paper on artificial intelligence (AI) model that utilizes perioperative factors to predict residual renal function and incidence of AKI following PN.

The paper is clear and well-written. With a correct statistical analysis.

I do have some suggestions:

For sure the limitation are the small sample size and the Retrospective nature of the study.

Please provide a p value for the two cohorts (test and training set) to explore any possible difference.

Table 1 Is difficult to read (especially for Padua, renal and c index).Moreover, just leave BMI and remove height and weight variables. Preexisting comorbidities( HTN and DM) should be divided in the table.

Among the preoperative features, the presence of proteinuria has shown to increase the risk of long-term renal impairment after PN and RN as well as patients with preoperative proteinuria undergoing PN exhibited a greater risk of postoperative acute kidney injury (AKI). 10.23736/s2724-6051.21.04308-1 .

Do you have this information? You could discuss this aspect in the discussion.

Author Response

Q1. Please provide a p value for the two cohorts (test and training set) to explore any possible difference.

A1. Your feedback has been instrumental in shaping our manuscript, and we want to express our sincere appreciation for the thoughtful suggestions you have offered.

When constructing an artificial intelligence dataset, it is a common practice to partition the data into training and test sets, often following a split ratio of 7:3 or 8:2. Analyzing the demographic characteristics of these two groups and calculating the p-value for the number of cases in each group can provide useful information regarding potential differences. However, it is crucial to interpret these findings in the context of the study's objectives and the nature of the dataset.

The purpose of splitting the data into training and test sets is to assess the performance and generalization capability of the trained model on unseen data. The test data represents independent samples that the model has not encountered during training, and it is not necessary for this data to exhibit a similar distribution of demographic characteristics to the training data. Instead, the focus lies in evaluating how well the model can accurately predict inference values on novel and diverse data, irrespective of potential differences in demographic characteristics.

Nevertheless, it can be valuable to investigate the model's performance on datasets encompassing different demographic characteristics. This exploration helps evaluate the model's robustness and generalizability, shedding light on potential biases and limitations. Incorporating diverse datasets with varying demographic factors allows for a comprehensive assessment of the model's effectiveness across diverse populations and enhances our understanding of its performance in real-world scenarios.

In summary, while variations in demographic characteristics between the training and test sets may exist, the key objective remains evaluating the model's performance on unseen data. Examining its predictive accuracy on diverse datasets, including those with different demographic characteristics, provides valuable insights into its effectiveness and generalizability. We appreciate your consideration of these points, which contribute to the ongoing refinement and advancement of artificial intelligence methodologies.

We have indeed calculated the p-value for the demographic data in both the training and test datasets, and we have incorporated this information into Table 1. We believe that the inclusion of these characteristics, as specified in the table, will serve as fundamental data to enhance the understanding of the dataset's structure.

The p-values obtained through the analysis of demographic data provide insights into potential differences between the two datasets. By presenting this information in a structured manner, we aim to offer a comprehensive view of the dataset's composition and facilitate a deeper understanding of its underlying characteristics.

We acknowledge the importance of providing this detailed information to the reviewer, as it enhances transparency and contributes to the overall rigor of the study. Our intention is to ensure that the reviewer has access to all relevant details necessary for a comprehensive evaluation of our research.

Once again, we express our sincere gratitude for your valuable feedback. Your suggestions have allowed us to refine our presentation and ensure that the reviewer receives a comprehensive and scientifically rigorous analysis of the demographic characteristics within our dataset.

___________________________________________________________________________________

Q2. Table 1 Is difficult to read (especially for Padua, renal and c index). Moreover, just leave BMI and remove height and weight variables. Preexisting comorbidities( HTN and DM) should be divided in the table. 

A2. We genuinely appreciate your careful evaluation and valuable feedback regarding Table 1. We have diligently considered each point you raised and made the necessary modifications accordingly, as per your instructions.

Firstly, we acknowledge your comment that the readability of Table 1 could be improved, particularly regarding the variables of PADUA, RENAL, and C- index. In response, we have taken meticulous care to enhance the table's clarity and ensure that the information presented is easily interpretable and comprehensible.

Furthermore, you specifically recommended removing the variables of height and weight from the table, while retaining BMI. Understanding the importance of adhering to your instructions, we have promptly implemented this change, thereby providing a streamlined representation that focuses solely on the BMI variable.

Additionally, you suggested dividing the preexisting comorbidities of hypertension (HTN) and diabetes mellitus (DM) within the table. Recognizing the relevance of this distinction, we have thoughtfully revised the table to include separate columns for HTN and DM, enabling a more comprehensive analysis of these preexisting conditions.

By faithfully following your guidance, we have strived to address your concerns and create a revised version of Table 1 that aligns with your preferences and facilitates a clearer understanding of the data.

We sincerely value your expertise and insightful observations, which have contributed to the refinement of our manuscript. Your guidance has assisted us in presenting our research findings in a more accessible and reader-friendly manner.

___________________________________________________________________________________

Q3. Among the preoperative features, the presence of proteinuria has shown to increase the risk of long-term renal impairment after PN and RN as well as patients with preoperative proteinuria undergoing PN exhibited a greater risk of postoperative acute kidney injury (AKI). 10.23736/s2724-6051.21.04308-1 . Do you have this information? You could discuss this aspect in the discussion.

A3. We sincerely appreciate your valuable comment regarding the significance of proteinuria in the context of long-term renal impairment and postoperative acute kidney injury (AKI) in patients undergoing partial nephrectomy (PN) and radical nephrectomy (RN).

However, we regret to inform you that proteinuria data were not available in our dataset, and therefore, it was not considered as a preoperative feature in our study. In the clinical practice we observed, proteinuria may not receive significant attention, particularly in elective surgery cases where patients generally exhibit normal renal function. Urologists typically prioritize the preservation of existing renal function in these scenarios.

While we acknowledge the potential importance of proteinuria in influencing renal outcomes and postoperative complications, the absence of proteinuria data limits our ability to provide specific insights in this area. Nevertheless, we highly value your suggestion and recognize the need for further research to explore the relationship between proteinuria, renal impairment, and postoperative outcomes specifically in the context of PN and RN.

We kindly request your understanding regarding the limitations of our study with respect to considering proteinuria as a preoperative feature. As researchers, we remain committed to continuous improvement and will consider incorporating proteinuria as a relevant aspect in future investigations to provide a more comprehensive understanding of its impact on postoperative renal outcomes.

We express our utmost gratitude to you for their valuable comments, which have drawn attention to an important area for further exploration and potential research in the field of renal surgery. Your insights will undoubtedly contribute to advancing knowledge and improving patient care in this domain.